# Quantitative Proteomics and Functional Characterization Reveal That Glutathione Peroxidases Act as Important Antioxidant Regulators in Mulberry Response to Drought Stress

**DOI:** 10.3390/plants11182350

**Published:** 2022-09-08

**Authors:** Minjuan Zhang, Wenqiang Li, Shuaijun Li, Junru Gao, Tiantian Gan, Qinying Li, Lijun Bao, Feng Jiao, Chao Su, Yonghua Qian

**Affiliations:** 1The Sericultural and Silk Research Institute, College of Animal Science and Technology, Northwest A&F University, Yangling 712100, China; 2State Key Laboratory of Crop Stress Biology in Arid Areas, College of Life Sciences, Northwest A&F University, Yangling 712100, China

**Keywords:** mulberry (*Morus alba* L.), drought stress, quantitative proteomics, antioxidant enzymes, glutathione peroxidase (GPX), reactive oxygen species (ROS)

## Abstract

Mulberry (*Morus alba* L.) has been an economically important food crop for the domesticated silkworm, *Bombyx mori*, in China for more than 5000 years. However, little is known about the mechanism underlying mulberry response to environmental stress. In this study, quantitative proteomics was applied to elucidate the molecular mechanism of drought response in mulberry. A total of 604 differentially expressed proteins (DEPs) were identified via LC-MS/MS. The proteomic profiles associated with antioxidant enzymes, especially five glutathione peroxidase (GPX) isoforms, as a scavenger of reactive oxygen species (ROS), were systematically increased in the drought-stressed mulberry. This was further confirmed by gene expression and enzymatic activity. Furthermore, overexpression of the GPX isoforms led to enhancements in both antioxidant system and ROS-scavenging capacity, and greater tolerance to drought stress in transgenic plants. Taken together, these results indicated that GPX-based antioxidant enzymes play an important role in modulating mulberry response to drought stress, and higher levels of GPX can improve drought tolerance through enhancing the capacity of the antioxidant system for ROS scavenging.

## 1. Introduction

The leaves of mulberry (*Morus alba* L.) have served as the unique feed for sericulture (silkworms) over thousands of years in China. However, mulberry trees are distributed all over the world. To some extent, this is due to strong environmental adaptability of the trees. In recent years, Chinese scientists of sericulture have proposed to expand the application of mulberry in ecological management and in diversified product development [1,2,3]. More and more mulberry trees have been planted in arid and semi-arid areas, especially in the northwestern of China, for ecological restoration [4,5,6]. Therefore, it is necessary to elucidate the mechanism underlying mulberry’s response and tolerance to drought stress.

Drought stress generally affects a vast range of morphological and physiological traits in plants, and reduces photosynthesis, leaf water potential, stem sap flow and stomatal conductance [7]. Physiologically, the common effect of drought stress is that the stress disturbs cellular water balance and cellular redox state, resulting in osmotic and oxidative stresses in plants [8,9]. Abscisic acid (ABA)-mediated signal pathways are believed to be the core of plant responses to drought stress, largely via osmotic adjustment [9,10,11]. Drought stress can lead to excessive generation and accumulation of reactive oxygen species (ROS) causing oxidative damage to cellular components [12,13]. ROS themselves behave as signaling molecules to trigger extensive biochemical and molecular effects that are essential for plant growth and development [14,15]. As ROS can be continuously produced in plant cells, the balance between the production and removal of ROS will be perturbed under stress conditions, evoking oxidative signaling to activate intrinsic mechanisms for ROS scavenging [16,17]. In plants, ROS scavenging medicated by superoxide dismutase (SOD), catalase (CAT), ascorbate peroxidase (APX) and glutathione peroxidase (GPX) constitutes the most important mechanism for plant antioxidant defense against abiotic stresses such as drought [18,19]. Therefore, activation and enhancement of the antioxidant system for the scavenging of excessive ROS are usually associated with increased tolerance of plants to abiotic stresses [16,17,20]. Plant GPXs are non-haeme thiol peroxidases that catalyze the reduction of H_2_O_2_ (or organic hydroperoxides) to water or the respective alcohols using reduced glutathione or thioredoxin [21,22]. GPXs were suggested to be a putative link between the glutathione-based and the thioredoxin-based detoxifying systems [23,24,25]. They possess some functional overlaps with peroxiredoxin (Prx) and glutathione transferases (GSTs), with respect to the maintenance of H_2_O_2_ homeostasis by elimination of peroxides, and are involved in the regulation of redox homeostasis by maintaining the thiol–disulfide balance [24,26,27]. There is increasing evidence to suggest that GPXs may play crucial roles in plant protection against both biotic and abiotic stresses [23,25,28].

Over the past few decades, remarkable achievements have been made in investigating the genetic and molecular mechanisms of plant abiotic-stress sensing and signaling [29]. However, due to slow reproduction, difficulties in genetic manipulation and larger plant size, etc., the genetic and molecular approaches extensively used in model plants are usually inefficient in investigating woody trees. The mechanism remains largely unknown in most tree species, such as mulberry. Transcriptomic and proteomic approaches could effectively identify target genes, proteins or signaling pathways that regulate plant growth, development and environmental stresses. Based on high-throughput proteomic technologies, a large number of proteins have been identified to be closely associated with stress responses in trees [30,31,32]. Isobaric tags for relative and absolute quantitation (iTRAQ), which has high sensitivity, good reproducibility, wide range, high-throughput analysis and other advantages, is a powerful proteomic technology to identify and quantify the levels of relevant sets of proteins [33].

In this study, iTRAQ-based quantitative proteomics was applied to identify differentially expressed proteins in the mulberry by experimentally withholding water. Our objective was to determine relevant proteins and/or pathways that are positively correlated with drought response in the mulberry. Combined with biochemical measurements and gene functional characterization, our results revealed that thiol-dependent antioxidant pathways, especially GPX isoforms, are most responsive to drought stress and overexpression of the GPX genes significantly increase the drought tolerance of transgenic plants. These findings will provide a foundation to screen drought-tolerant germplasm in mulberry.

## 2. Results

### 2.1. Physiological and Biochemical Characterization of the Mulberry under Drought Stress

The mulberry plants were drought stressed by withholding water for 11 days. Soil water content in the pots growing drought-stressed plants was reduced about 4.6% after 6 days of withholding water and further reduced to below the detection limit of the instrument after 8 days of withholding water, whereas soil water content in the pots of non-stressed plants was not less than 22%. After 11 days of withholding water, the top leaves (1st to 3rd leaf) in each plant exhibited visible dehydration, the middle leaves (5th to 7th leaf) exhibited severe dehydration, and the bottom leaves were extremely dehydrated, yellowed and even fell (Figure 1A). Compared with ~90% relative water content (RWC) in non-stressed plants, the RWC in drought-stressed plants were about 76.1, 62 and 30.2% in top, middle and bottom leaves, respectively (Figure 1B). Compared with that in non-stressed plants, drought stress caused significant reductions in the total chlorophyll content (Figure 1C) and the soluble sugar content (Figure 1D), respectively. The malondialdehyde and free proline contents were significantly increased under drought stress (Figure 1E,F). The content of soluble protein was not significantly changed under the drought stress (Figure 1G).

### 2.2. Quantitative Proteomic Analysis of the Mulberry under Drought Stress

To investigate the changes of mulberry proteome in response to drought stress, iTRAQ-based quantitative proteomics was used. The proteomes were labeled with iTRAQ reagent and then quantified by LC-MS/MS. The length for most of the peptides was distributed between 7 and 21 amino acids (Figure 2A), which agrees with the property of tryptic peptides and means that sample preparation reaches standard. Mass errors, meaning the mass accuracy of MS data, were checked for all the identified peptides. Distribution of mass errors in 0~10 also reached standard and were sufficient for the further analyses (Figure 2B). Principal component analysis (PCA) showed that the datasets from both non-stress and drought-stress samples clustered within their own functional group (Figure 2C). A summary of the MS/MS spectrum analysis, including total spectrums, matched spectrums, peptides and unique peptides, are shown (Figure 2D). Based on the MS/MS data, 2495 proteins were identified (Figure 2D), of which 2058 proteins were quantified in the drought-stress and non-stress samples of mulberry (Appendix A).

Statistical analysis revealed that 604 proteins changed in abundance significantly (Appendix A); these were defined as differential expressed proteins (DEPs). In these DEPs, 278 proteins were up-regulated, and 326 proteins were down-regulated (Appendix A). According to the results of euKaryotic Ortholog Groups (KOG) of protein comparison, the DEPs were annotated to 22 classifications, including posttranslational modification, protein turnover, chaperones, translation, ribosomal structure, and biogenesis, etc. (Figure 3A). Gene ontology (GO) enrichment analysis revealed that molecular functions of the DEPs were significantly enriched in terms of carbohydrate binding, hydrolase activity, disulfide oxidoreductase activity, polysaccharide binding, polygalacturonase inhibitor activity, antioxidant activity, molecular function regulator, and glutathione disulfide oxidoreductase activity (Figure 3B). In cellular component category, the DEPs were significantly enriched in terms of anchored component of membrane, anchored component of plasma membrane, and intrinsic component of plasma membrane, etc. (Figure 3B). In biological process category, the DEPs were significantly enriched in terms of positive regulation of signal transduction, response to hydrogen peroxide, cellular oxidant detoxification, etc. (Figure 3B).

Kyoto Encyclopedia of Genes and Genomes (KEGG) pathway enrichment analysis revealed that 9 KEGG pathways, including protein processing in endoplasmic reticulum (ER), glutathione metabolism, galactose metabolism, glycerolipid metabolism, MAPK signaling pathway, arachidonic acid metabolism, etc., were significantly enriched (Figure 3C). As shown in KEGG map (Appendix A), several key steps in glutathione metabolism pathway were affected by drought stress. It implied that glutathione peroxidases (GPXs: EC 1.11.1.9 and EC 1.11.1.12) were significantly up-regulated, while L-ascorbate peroxidase (APX: EC 1.11.1.11) was significantly down-regulated (Appendix A). The result of InterPro protein domain enrichment indicated that alcohol dehydrogenase GroES-like domain, glycosyl hhdrolases family 17, Glutaredoxin, zinc-binding dehydrogenase and thaumatin family, etc., were significantly enriched (Figure 3D).

### 2.3. Abundance of Antioxidant Enzymes Especially GPX Isoforms Were Systematically Increased in the Mulberry under Drought Stress

Proteomic analysis showed that 43 DEPs are associated with cellular redox and antioxidant system, accounting for 7% of the total number of DEPs (Appendix A). A great number of enzymatic and non-enzymatic antioxidants were changed in protein abundance under drought stress (Appendix A). It is showed that the thiol-dependent antioxidants, including four peroxiredoxins (W9QVC2, W9SEV0, W9QVC2, W9SEV0), five GPXs (W9QHE0, W9QT41, W9QH65, W9RT74 and W9SDB3), five thioredoxins (W9SDD7, W9SW93, W9RMD9, W9R4T1 and W9R4T1), five glutaredoxins (W9S7L1, W9QZP8, W9SBA6, W9SCK7, W9SCK7), two glutathione S-transferases (W9RCX8 and W9S168) and a peroxidase (W9SE23), were all up-regulated under drought stress (Table 1). A L-ascorbate oxidase (W9RQI7) and a superoxide dismutase SodC (W9SBU2) were significantly down-regulated in protein abundance (Table 1). The results implied that thiol-dependent antioxidants, especially the GPX isoforms, were significantly induced in the mulberry under drought stress.

Analysis of the mulberry genome has identified 6 typical GPX isoform genes, the *MaGPX1*~*6* [34]. A phylogenetic tree of GPX isoforms from mulberry (MaGPXs), *Arabidopsis* (AtGPXs) and rice (OsGPXs) were shown and indicated the similarities of these GPX members (Appendix A). We further investigated the gene expression of *MaGPXs* in mulberry Neo-Ichinose under drought stress. Compared with the non-stressed mulberry plants, the drought stress induces about 3~8 folds increases in mRNA expression in *MaGPX1*, *MaGPX2*, *MaGPX3* and *MaGPX5* (Figure 4A). This is consistent with the proteomic data that the protein abundance of GPX isoforms was significantly increased in mulberry Neo-Ichinose under drought stress. According to the proteomic result, the proteins abundance of MaGPX1 (W9QH65), MaGPX2 (W9RT74), MaGPX3 (W9QHE0), MaGPX4 (W9SDB3) and MaGPX5 (W9QT41) were increased, while MaGPX6 (W9QS90) was not changed under drought stress (Figure 4B). These results implied that drought stress increases the abundance of GPX proteins correlated with increased expression of GPX genes.

### 2.4. Enzymatic Activity of GPX Was Increased in the Mulberry under Drought Stress

We determined the enzymatic activities of antioxidants including SOD, APX, CAT, GPX, peroxidase (POD), glutathione reductase (GR) and glutathione S-transferase (GST) in the mulberry. The enzymatic activities of SOD, CAT and peroxidase (POD) were not significantly changed after drought stress (Figure 5A–C). The enzymatic activity of APX was decreased under drought stress (Figure 5D), implying that the ascorbate–glutathione (AsA-GSH) cycle was depressed under drought stress. The enzymatic activity of GPX was significantly increased (Figure 5E), and the content of GSH was also increased (Figure 5F), indicating the activation of the GPX cycle in the mulberry under drought stress. The enzymatic activity of glutathione reductase (GR) was not changed (Figure 5G), but the activity of glutathione S-transferase (GST) was significantly decreased under drought stress (Figure 5H). Moreover, the H_2_O_2_ content was not significantly changed after drought stress (Figure 5I). These results demonstrated that drought stress may enhance the GPX-based antioxidant system in the mulberry.

### 2.5. Ectopic Overexpression of Mulberry GPX Isoforms Confered Drought Resistance in Transgenic Arabidopsis

To further investigate the mulberry GPX isoforms, we produced transgenic *Arabidopsis* by overexpressing the six GPX genes, *MaGPX1* to *MaGPX6*, respectively. A schematic diagram of the MaGPXs expression vectors are shown in Figure 6A. To obtain ectopic overexpression lines, hygromycin-resistant transgenic plants were identified by GUS staining (Appendix A). Overexpressions of the MaGPX genes in transgenic plants were confirmed by qRT-PCR (Appendix A). Before drought stress, no phenotypic difference was observed between the control (containing empty vector) and the overexpression transgenic plants (Figure 6B, left). After 14 days of withholding water, the control, OE-MaGPX1, OE-MaGPX2, OE-MaGPX4 and OE-MaGPX6 transgenic lines were absolutely wilting and dried, but the OE-MaGPX3 and OE-MaGPX5 transgenic lines were still alive (Figure 6B, middle). After re-watering for 5 days, the OE-MaGPX3 and OE-MaGPX5 plants recovered to normal growth, but the other transgenic lines and the control were not recovered from drought stress (Figure 6B, right). The result revealed that the overexpression of mulberry GPX isoforms *MaGPX3* and *MaGPX5* resulted in an increased tolerance of transgenic plants to drought stress.

We further evaluated drought tolerance of OE-MaGPX3 and OE-MaGPX5 transgenic lines at the flowering stage (Figure 6C, left). After 12 days of withholding water, the control displayed a severe wilting and dry phenotype, but the OE-MaGPX3 and OE-MaGPX5 lines showed a relatively mild phenotype (Figure 6C, middle). A week after re-watering, all the OE-MaGPX3 and OE-MaGPX5 transgenic plants were recovered, but all the controls were not able to recover (Figure 6C, right). These data revealed that the ectopic overexpression of *MaGPX3* and *MaGPX5* leads to the greater resistance of transgenic plants to drought stress.

We measured relative water content (RWC) and rate of water loss (RWL) in the control, OE-MaGPX3 and OE-MaGPX5 transgenic lines. It was showed that OE-MaGPX3 and OE-MaGPX5 transgenic plants had significantly higher RWC (Figure 6D) and lower RWL (Figure 6E) as compared with that of control under non-stress condition. The drought resistance of the transgenic lines was also confirmed by a mannitol-induced drought-stress experiment (Appendix A). The mannitol treatment caused a significant decrease in seedling length in the controls, but no decrease in the OE-MaGPX3 and OE-MaGPX5 transgenic lines (Appendix A).

### 2.6. Ectopic Overexpression of Mulberry GPX Isoforms Improve Both Antioxidant Enzymatic Activities and ROS Scavenging in Transgenic Arabidopsis under Drought Stress

To investigate why overexpression of *MaGPX3* and *MaGPX5* increases the tolerance of transgenic plants to drought stress, we analyzed the activities of the antioxidant system and ROS accumulation in transgenic plants. Under both non-stress and drought-stress conditions, GPX enzymatic activities were significantly higher in the OE-MaGPX3 and OE-MaGPX5 transgenic lines than in the control (Figure 7A). Compared with that of non-stressed plants, GPX activities showed 311% and 300% increases in OE-MaGPX3 and OE-MaGPX5 transgenic lines, but a 109% increase in the control under drought stress (Figure 7A). These results revealed that overexpression of *MaGPX3* and *MaGPX5* may strongly increase enzymatic activity of GPX in transgenic plants under non-stress and drought-stress conditions.

Enzymatic activity of APX was significantly higher in OE-MaGPX3 and OE-MaGPX5 than in the control under both non-stress and drought-stress conditions (Figure 7B). Under non-stress conditions, the enzymatic activities of SOD, CAT, POD, GR and GST were not significantly higher in the OE-MaGPX3 or OE-MaGPX5 transgenic lines than in the control; however, under drought-stress condition, the enzymatic activities of SOD, CAT, POD, GR and GST were significantly higher in the OE-MaGPX3 and OE-MaGPX5 transgenic lines than in the control (Figure 7C–G). These results revealed that overexpression of *MaGPX3* and *MaGPX5* not only increases the enzymatic activity of GPX, but also improves the activities of other antioxidant system enzymes in transgenic plants exposed to drought stress.

ROS accumulation was then investigated in the control, OE-MaGPX3 and OE-MaGPX5 transgenic plants. The content of hydrogen peroxide (H_2_O_2_) was significantly lower in OE-MaGPX3 and OE-MaGPX5 transgenic lines than in the control under both non-stress and drought-stress conditions (Figure 7H). It was showed that drought stress caused a larger increase in hydrogen peroxide content in the control, but a smaller increase in hydrogen peroxide content in the overexpression lines (Figure 7H). Although drought stress caused significant increases in hydrogen peroxide content in the control (105% increases), OE-MaGPX3 (69% increase) and OE-MaGPX5 (50% increase), the hydrogen peroxide levels in drought-treated overexpression lines were as low as that in the non-stressed control plants (Figure 7H), indicating enhanced ROS (hydrogen peroxide) scavenging by overexpressing MaGPX3 and MaGPX5.

DAB and NBT histochemical staining experiments were further performed to investigate ROS (hydrogen peroxide and superoxide) accumulation. Under non-stress conditions, no visible difference in the DAB and NBT staining of leaves was observed between the overexpression transgenic plants and the control plants (Figure 7I,J), indicating that the levels of hydrogen peroxide and superoxide were not different between the overexpression plants and control plants. Under drought-stress conditions, however, the overexpression lines were significantly less stained by DAB and NBT as compared with that of the control (Figure 7I,J), indicating that overexpression of *MaGPX3* and *MaGPX5* led to less accumulation of hydrogen peroxide and superoxide in transgenic plants under drought stress. The results revealed that the overexpression of *MaGPX3* and *MaGPX5* could increase antioxidant enzymatic activities and lead to less ROS accumulation in drought-stressed plants.

## 3. Discussion

For most plants, drought stress can affect a vast range of morphological and physiological traits, such as reductions in photosynthesis, leaf water potential, stem sap flow and stomatal conductance [7]. In this study, compared with non-stressed plants, leaf RWC, total chlorophyll content, and soluble sugar content were decreased, but malondialdehyde and proline contents were increased in the stressed mulberry plants, indicating that drought stress reduces water transport, photosynthesis and carbohydrate metabolism, and causes cellular dehydration leading to osmotic and oxidative stress in mulberry. This is consistent with previous studies in other tree species, i.e., *Maclura pomifera* [35], *Prunus sargentii* and *Larix kaempferi* [36].

In this study, iTRAQ-based quantitative proteomics was used to investigate the molecular mechanism of mulberry’s response to drought stress. We identified 43 DEPs associated with cellular redox, including many antioxidant system enzymes that are responsible for ROS scavenging or ROS-scavenger recovering (Appendix A). The abundance of antioxidant system enzymes such as SOD, GPX, Trx, Grx, GST and APX were extensively changed in the mulberry under drought stress (Table 1). Since most of these antioxidant proteins were upregulated by drought stress, the results suggest that drought stress activates the antioxidant system in the mulberry.

Under abiotic stress, especially environmental stress (i.e., drought), a plant produces more ROS than it needs [18]; however, the plant also produces more antioxidants, flavonoids, and secondary metabolites which play the role in protecting the plant for detoxifying ROS and maintaining protein and amino-acid stabilization. It is generally accepted that ROS scavenging or detoxifying medicated by antioxidant system such as SOD, CAT, APX and GPX constitutes the most important mechanism of plant defense against abiotic stresses [17,18,19]. As important ROS scavengers, SOD converts superoxide into hydrogen peroxide (H_2_O_2_); APX, GPX, and CAT may detoxify H_2_O_2_ to H_2_O [37]. APX catalyzes the conversion of H_2_O_2_ into H_2_O by the oxidation of ascorbate to monodehydroascorbate (MDA); however, GPX detoxifies H_2_O_2_ into H_2_O by oxidation of glutathione (GSH) to oxidized GSH [16]. In some cases, it is considered that plant GPX isoforms use thioredoxin (Trx) rather than GSH in the reduction of H_2_O_2_ and lipid hydroperoxides [23,26]. In fact, the levels of antioxidant system enzymes, i.e., APX and GPX could be differentially regulated in plant responses to environmental stresses and/or during normal growth [20,27,38,39,40,41,42]. In this study, drought stress led to a significant decrease in APX activity as well as reduced APX protein abundance in the mulberry. This may imply that APX-mediated ROS scavenging was depressed in the mulberry under drought stress. It also suggests that the APX-mediated ROS scavenging pathway plays a less important role in the mulberry-plant defense against drought stress.

The drought stress resulted in significant increases in GPX gene expression, GPX isoenzymes abundance as well as GPX enzymatic activity in the mulberry under drought stress. This is consistent with some previous studies that GPX was up-regulated in response to environmental stresses [22,43,44]. Given that the mulberry genome encodes 6 GPX members, the results indicated that mulberry GPX isoenzymes, rather than other antioxidant system enzymes (ie. APX), were extensively induced in mulberry under drought stress. We suggest that GPX could play a prominent role in regulating antioxidant defense against drought stress in the mulberry.

Some investigations suggest that the subcellular compartmentalization of antioxidant system enzymes (i.e., GPX isoenzymes) is very important for plant cells to modulate ROS concentration and protect plants from oxidative stress [23,25,40,42]. Our previous study indicated that mulberry GPX members may be present in different subcellular compartments [34]. In this study, five mulberry GPX isoforms were increased in mRNA expression, protein abundance and enzymatic activity, suggesting that mulberry GPX isoforms were uniformly enhanced in different subcellular compartments to maintain ROS level during drought stress. This consideration is partly supported by the result that drought stress only induced a slight increase in H_2_O_2_ level (no significance) in the mulberry Neo-Ichinose (Figure 5I). In this respect, it seems that the mulberry variety Neo-Ichinose, as a drought-tolerant variety, possess an efficient antioxidant system that may alleviate drought-induced ROS overproduction in some degree.

As a major family of the thiol-based ROS scavenging enzymes, GPX isoforms were initially described as catalyzing the reduction of hydrogen peroxide and lipid peroxides using reduced GSH as electron donor [45,46]. The overexpression of *MaGPX3* and *MaGPX5* led to less accumulation of ROS in drought-stressed transgenic plants, indicating that the GPX isoforms play a positive role in detoxifying ROS. There is increasing evidence that GPX isoforms are not only regulated by various abiotic stresses, but also involved in modulating plant tolerance to environmental stresses [24,27,47,48]. Some previous studies suggest that transgenic plants with additional copies of GPX have higher resistance to stresses; conversely, the plants with the knockout of individual GPX genes are less tolerant to the stress [42]. For example, the overexpression of *Synechocystis* GPX in *Arabidopsis* can reinforce the tolerance of transgenic plants to oxidative, chilling, salinity and drought stresses [49]. Overexpression of *Rhodiola crenulata GPX5* affects the regulation of multiple biochemical pathways and increases the drought tolerance in *Salvia miltiorrhiza* [50]. These investigations are consistent with our result that the overexpression of mulberry *MaGPX3* and *MaGPX5* increase drought tolerance in plants. This suggests that *MaGPX3* and *MaGPX5* may play a positive role in modulating mulberry’s tolerance to drought stress. Taken together, we considered that a higher abundance of GPX isoforms is associated with greater antioxidant abilities and higher tolerance to drought stress. Levels of GPX gene expression and enzymatic activity may be used as an index to evaluate drought-tolerant germplasm in mulberry.

## 4. Materials and Methods

### 4.1. Plant Materials and Growth Conditions

The Neo-Ichinose, a drought- and salt-tolerant mulberry variety (*Morus alba* L.), originally introduced from Japan, was obtained from the mulberry germplasm resources pool in the Sericultural and Silk Research Institute of Northwest A&F University (Yangling, Shaanxi, China). One-year-old grafted seedlings of mulberry were transplanted into plastic pots (45 cm diameter and 30 cm height) for 2 weeks. One plant was planted in a pot with 15 kg of soil. All plants were watered with 1000 mL of water every two days prior to the initiation of drought stress. The plants were grown in a greenhouse with a 14-hour-light (34 °C)/10-hour-dark (28 °C) cycle and 40–50% relative humidity.

Seeds of overexpression transgenic *Arabidopsis thaliana* lines and control (Columbia-0 background) were germinated on 1/2 MS agar medium for 5 days. After germination, seedlings with equal size were selected and transferred into square pots with seedling medium (1:1 mixture of peat and vermiculite). The seedlings were incubated with 16 h light/8 h dark cycles at 22 °C in a growth chamber.

### 4.2. Drought Treatments

Mulberry plants were treated by withholding water for 11 days, while non-stressed plants were still watered with 1000 mL of water for each pot every two days. To evaluate the drought degree, soil volumetric water content was measured using a soil moisture content analyzer (TZS-IIW, Hangzhou TOP Instrument Co., Ltd., Hangzhou, China) every two days. After withholding of water for 11 days, the bottom leaves of plant were extremely dehydrated and even fell. For proteomic, physiological and antioxidant enzymatic measurements, the second and third leaves from top of plant were collected and immediately frozen in liquid nitrogen and stored at −80 °C until use. Biochemical measurements were performed with six biological replicates, and one pot was sampled as one replicate. To evaluate drought tolerance of transgenic *Arabidopsis* plants with overexpression of mulberry GPX genes, 3-week-old *Arabidopsis* plants (T_3_ generation) were subjected to drought stress by withholding water for 14 days, and then followed by re-watering for 5 days. To further confirm drought tolerance, 40-day-old *Arabidopsis* plants were subjected to drought stress by withholding water for 12 days and then followed by recovery for 7 days. For measurement of antioxidant system enzymes, the seedlings of *Arabidopsis* plants were subjected to drought stress by withholding water for 6 days when the control plants were lightly wilting. Leaves from non-treated and drought-treated plants were sampled and immediately frozen in liquid nitrogen and stored at −80 °C until use. Mannitol-induced drought-stress experiment was applied by germinating of seeds on 1/2 MS agar medium supplemented with 0 or 200 mmol L^−1^ mannitol. After 10 days of culture, the lengths of seedlings were measured.

### 4.3. Biochemical Measurements

Contents of malondialdehyde, free proline, soluble sugar, soluble proteins and total chlorophyll were determined as described previously [51,52]. Relative water content (RWC) and rate of water loss (RWL) were measured as described by previous protocols [53]. To determine RWC, three leaf weights were taken: W1, immediately after leaf excision; W2, after wiping off water following a 4 h water saturation of excised leaves at room temperature; and W0, after subsequently drying leaves for 12 h at 70 °C. RWC (%) = (W1 − W0)/(W2 − W0) × 100%. To determine RWL, detached leaves were initially soaked in water for 1 h, after which excess water was removed and, thereafter, leaves were weighed every 30 min. For antioxidant enzymatic activities, the leaves were ground in liquid nitrogen and assayed for activities of antioxidant enzymes including GPX, APX, SOD, CAT, POD, GR, GST and content of GSH by commercially available antioxidant enzyme kits (Nanjing Jiancheng Bioengineering Institute, Nanjing, China) according to manufacturer’s instructions. The detailed experimental procedures for antioxidant enzyme activities are available in Appendix A.

### 4.4. iTRAQ-Based Quantitative Proteomics and Bioinformatic Analysis

Isobaric tags for relative and absolute quantitation (iTRAQ)-based quantitative proteomics was performed using a customer service by Jingjie PTM-Biolabs Co., Ltd. (Hangzhou, China). In brief, total proteins were extracted from mulberry leaves and then subjected to trypsin digestion. After trypsin digestion, iTRAQ labeling for three drought-treated samples and three control samples were performed according to the manufacturer’s protocol with iTRAQ Reagent-8 Plex Multiplex Kit (AB SCIEX, Framingham, MA, USA). iTRAQ-labeled peptides were fractionated by HPLC and then analyzed by LC-MS/MS (Thermo Scientific Q Exactive Plus, Waltham, MA, USA). MS data were analyzed using MaxQuant software (version 1.5.2.8, https://www.maxquant.org/, (accessed on 2018)), followed by database searching and bioinformatics analysis. Tandem mass spectra were searched against UniProt database concatenated with reverse decoy database. Trypsin/P was specified as cleavage enzyme allowing up to 2 missing cleavages, 4 modifications per peptide and 5 charges. Mass error was set to 10 ppm for precursor ions and 0.02 Da for fragment ions. Carbamidomethylation on Cys was specified as fixed modification and oxidation on Met and acetylation on protein N-terminal were specified as variable modifications. False discovery rate (FDR) threshold was specified at 1% and ion score was set >20. Minimum peptide length was set at 6. All the other parameters were set to default values. The relative quantitation of proteins was performed according to the quantitative ratio between drought-treated and control samples. A *p* value < 0.05 from *t*-tests and a fold change >1.5 or <0.67 (1/1.5) were set as the thresholds for differentially expressed protein (DEP). Gene ontology (GO; http://www.geneontology.org/, (accessed on 2019)) and Kyoto Encyclopedia of Genes and Genomes (KEGG; http://www.genome.jp/kegg/, (accessed on 2019)) databases were used to classify and group the identified proteins, and Fisher’s exact tests were used to define significantly enriched GO terms and pathways using all the DEPs as target sets. Protein domain annotation was performed using InterProScan software (http://www.ebi.ac.uk/interpro/, (accessed on 2019)). A detailed experimental procedure for iTRAQ-based quantitative proteomics is available in Appendix A.

### 4.5. Ectopic Overexpression of Mulberry GPX Genes in Arabidopsis

The open reading frame (ORF) sequence of *MaGPX1* to *MaGPX6* were subcloned into a binary plasmid *pCAMBIA1301,* respectively, under the control of CaMV35S promoter as we described previously [34]. The recombinant constructs and *pCAMBIA1301* vector (control) were introduced into *Agrobacterium* strain GV3101, respectively, and then transformed into *Arabidopsis thaliana* Col-0 using the floral dip method. To identify the overexpression lines, T_1_ transgenic *Arabidopsis* plants were screened on 1/2 MS agar plates supplemented with hygromycin (30 mg L^−1^) and then used for GUS staining analysis. To determine expression levels of MaGPX genes, positive T_1_ plants were further subjected to RNA extraction and quantitative RT-PCR. Hygromycin-resistant screening was continued for the T_2_ and T_3_ transgenic lines.

### 4.6. RNA Extraction and Quantitative RT-PCR

Total RNAs were isolated from mulberry and *Arabidopsis* leaves using RNA Extraction Kit (TaKaRa, Dalian, China) according to the manufacturer’s protocol. The extracted RNAs were treated with RNase-free DNase I (TaKaRa, Dalian, China) for eliminating genomic DNA. The first strand of cDNA was synthesized using the PrimeScript RT Reagent Kit (TaKaRa, Dalian, China). Quantitative RT-PCR (qRT-PCR) was performed with the SYBR^®^ Premix Ex Taq™ II (TaKaRa, Dalian, China) on the Bio-Rad CFX96 Real-Time PCR System. The relative expression levels of MnGPX genes were obtained by normalization to internal reference gene *AtACT1* in transgenic Arabidopsis or *MaACT1* in mulberry and calculated using the 2^–ΔΔCt^ method. The primers used for qRT-PCR analysis are listed in Appendix A.

### 4.7. ROS Measurement and Histochemical Staining

ROS is a collective term that includes both oxygen free radicals (i.e., superoxide) and nonradicals (i.e., hydrogen peroxide). Hydrogen peroxide (H_2_O_2_) content was assessed using a commercial hydrogen peroxide assay kit (Nanjing Jiancheng Bioengineering Institute, Nanjing, China) according to manufacturer’s instructions. H_2_O_2_ bound to molybdenum acid forms a complex, measured at 405 nm, from which the content of H_2_O_2_ was calculated. The visualization of H_2_O_2_ and superoxide anion (O_2_^.-^) in leaves were detected by 3,3′-diaminobenzidine (DAB) and nitroblue tetrazolium chloride (NBT) histochemical staining, respectively [54].

## 5. Conclusions

In this study, quantitative proteomics combined with biochemical analyses and gene functional characterization were used to investigate the mechanism of mulberry tree’s response to drought stress. Drought stress reduces water transport, photosynthesis, and carbohydrate metabolism in the mulberry, and causes cellular dehydration, and osmotic and oxidative stress. Quantitative proteomic and biochemical analyses reveal that drought stress causes extensive change in antioxidant-system enzymes, especially a systematic upregulation of five GPX isoforms and increased antioxidant activity in the mulberry, suggesting a prominent role of GPX in regulating antioxidant defense against drought stress. More importantly, overexpression of the GPX members leads to greater tolerance to drought stress, enhanced capacity of antioxidant system and less accumulation of ROS in transgenic plants. Our results suggest that a higher abundance of GPX isoforms is associated with greater antioxidant abilities and higher tolerance to drought stress in mulberry. These findings will provide a new approach for identifying drought-tolerant germplasm in mulberry.

## Figures and Tables

**Figure 1 plants-11-02350-f001:**
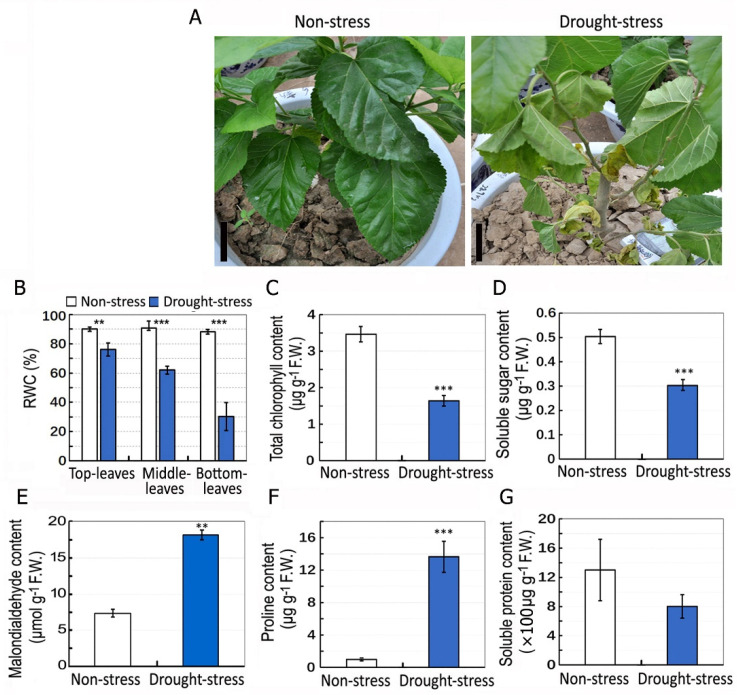
Phenotypes and physiological analyses of the mulberry under drought stress. (**A**) The phenotypes in control (non-stress) and drought-stressed plants. Mulberry plants were subjected to drought stress by withholding water for 11 days, while the control plants were irrigated regularly. Bars = 8 cm. (**B**) Relative water content (RWC) in top leaves (1st to 3rd leaf), middle leaves (5th to 7th leaf) and bottom leaves (9th to 12th leaf) of control and drought-stressed plants. (**C**–**G**) Total chlorophyll content (**C**), soluble sugar content (**D**), malondialdehyde content (**E**), proline content (**F**) and soluble proteins (**G**) in top leaves of control and drought-stressed plants. Data are means ± SD with five biological replicates. Asterisks represent statistically significant differences between control and drought-treated plants by Student’s *t*-test (** *p* < 0.01, *** *p* < 0.001).

**Figure 2 plants-11-02350-f002:**
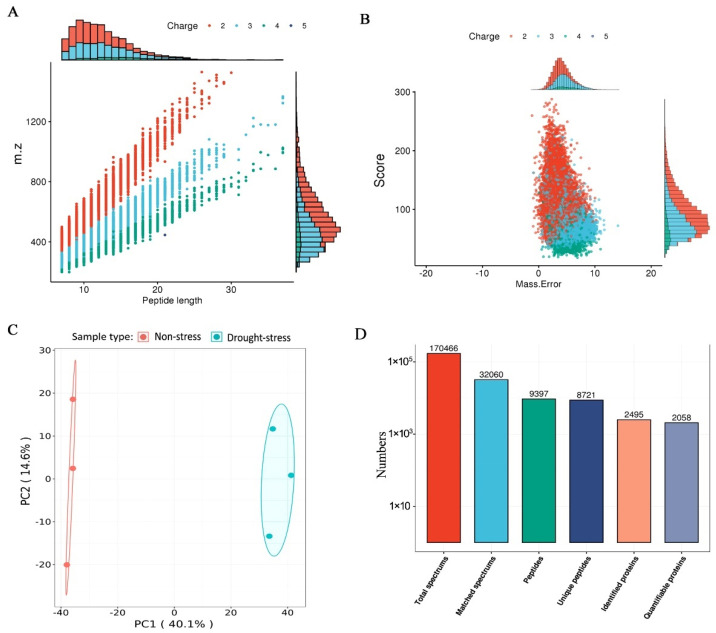
iTRAQ-based quantitative proteomics of the mulberry under drought stress. (**A**–**C**) The distribution of peptide length (**A**), the peptide mass tolerance (**B**) and principal component scatter plot analysis (**C**) for all peptides. (**D**) The MS/MS spectrum database search analysis summary.

**Figure 3 plants-11-02350-f003:**
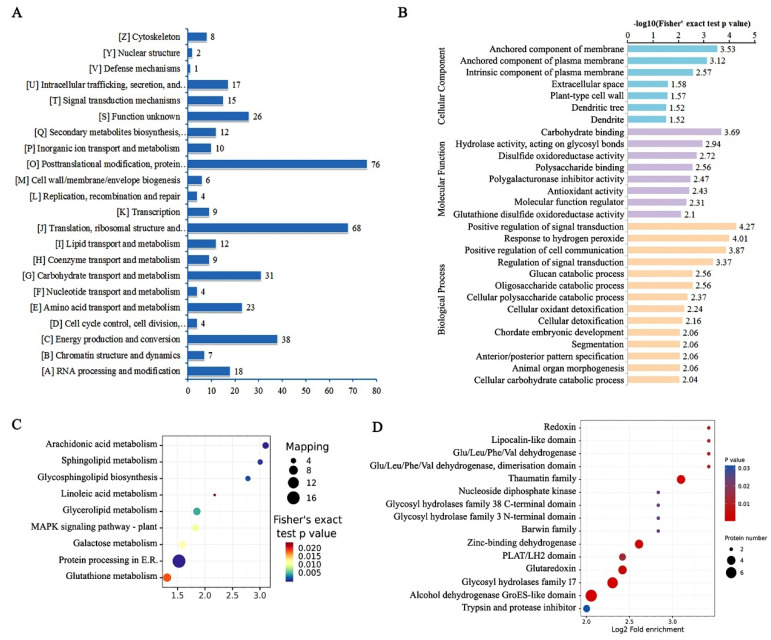
Functional classification and enrichment analysis of the differentially expressed proteins (DEPs) from iTRAQ-based quantitative proteomics. (**A**–**D**) EuKaryotic Ortholog Groups classification (**A**), Gene ontology enrichment (**B**), KEGG pathway enrichment (**C**) and InterPro protein domain enrichment (**D**).

**Figure 4 plants-11-02350-f004:**
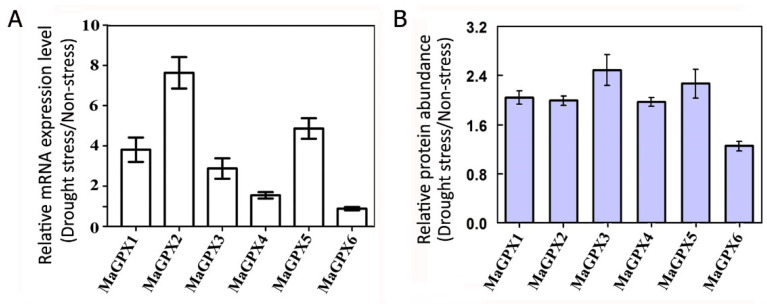
The expression levels of MaGPXs genes and proteins in the mulberry under drought stress. (**A**) Relative mRNA levels of *MaGPX1*, *MaGPX2*, *MaGPX3*, *MaGPX4*, *MaGPX5*, and *MaGPX6* in mulberry seedlings under drought stress. The data represents the ratio between drought-treated plants and non-treated plants. qRT-PCR was performed with three biological replicates each with three technical replicates. (**B**) Fold change in MaGPXs protein levels between drought stress and non-stress. The data are selected from iTRAQ-based quantitative proteomics.

**Figure 5 plants-11-02350-f005:**
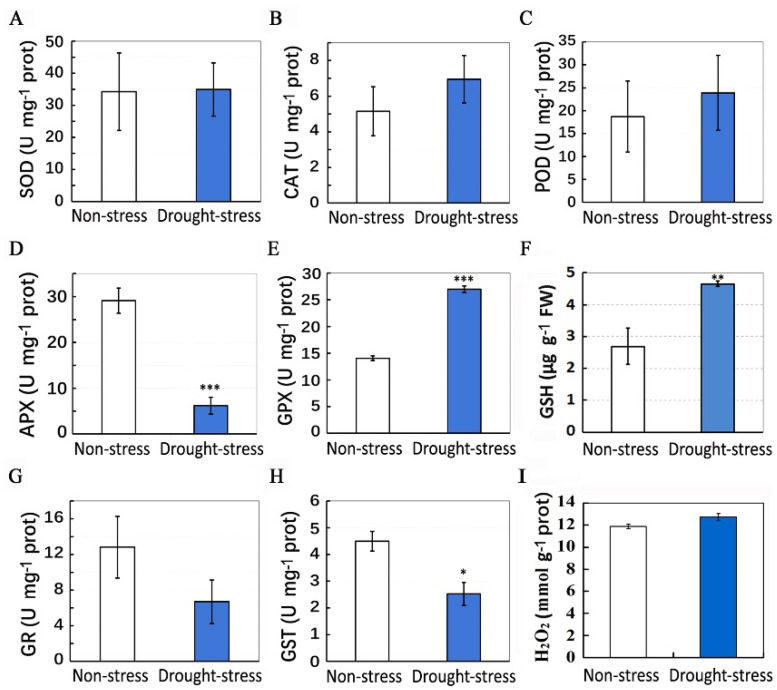
Activities of antioxidant enzymes and the H_2_O_2_ content in control (non-stress) and drought-stressed plants. (**A**–**I**) Enzymatic activities of SOD (**A**), CAT (**B**), POD (**C**), APX (**D**), GPX (**E**), GR (**G**), GST (**H**) and GSH content (**F**), and the H_2_O_2_ content (**I**) in top leaves of control and drought-stressed mulberry plants. Data are means ± SD with five biological replicates. Asterisks represent statistically significant differences between control and drought-stressed plants by Student’s *t*-test (* *p* < 0.05, ** *p* < 0.01, *** *p* < 0.001).

**Figure 6 plants-11-02350-f006:**
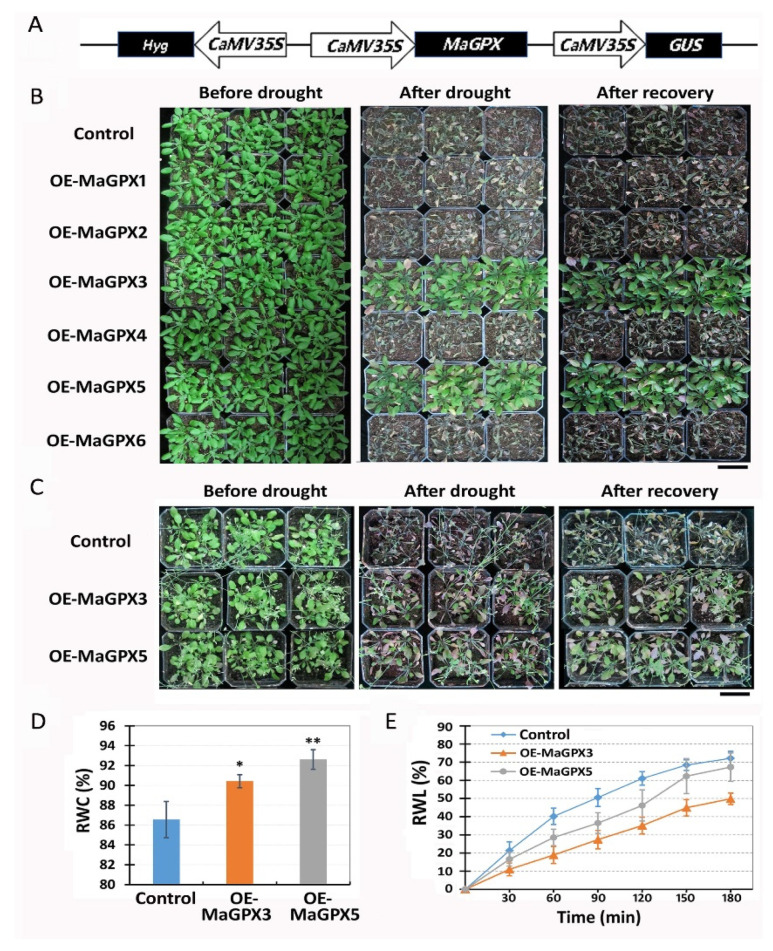
Overexpression of mulberry *MaGPX3* and *MaGPX5* enhances drought tolerance in transgenic Arabidopsis. (**A**) Drought tolerance of transgenic Arabidopsis overexpressing six mulberry GPX genes, *MaGPX1*, *MaGPX2*, *MaGPX3*, *MaGPX4*, *MaGPX5* and *MaGPX6* at the seedling stage. Phenotypes of control and the transgenic lines (OE-MaGPXs) before drought (3-week-old plants), after withholding water for 14 days, and after recovery for 5 days are shown, respectively. Bar = 4 cm. (**B**) Drought tolerance of the control, OE-MaGPX3 and OE-MaGPX5 transgenic lines at the flowering stage. The phenotypes of control and transgenic lines before drought (40-day-old plants), after withholding water for 12 days, and after recovery for 7 days are shown, respectively. Three independent T_3_ lines were used in each case. Bar = 4 cm. (**C**) Relative water content (RWC) of leaves of the control, OE-MaGPX3 and OE-MaGPX5 transgenic plants in non-stress condition. (**D**) Rate of water loss (RWL) of leaves of the control, OE-MaGPX3 and OE-MaGPX5 transgenic plants in non-stress condition. Data are means ± SD with six biological replicates. Asterisks represent statistically significant differences between control and transgenic plants by Student’s *t*-test (* *p* < 0.05, ** *p* < 0.01).

**Figure 7 plants-11-02350-f007:**
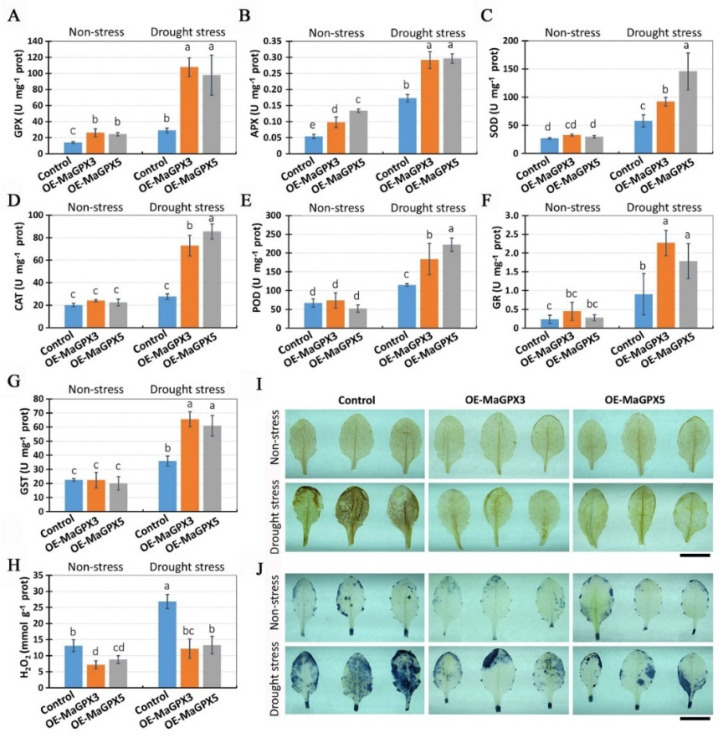
Antioxidant enzyme activities and ROS levels in the control, OE-MaGPX3 and OE-MaGPX5 transgenic Arabidopsis plants. (**A**–**G**) Enzymatic activities of GPX (**A**), APX (**B**), SOD (**C**), CAT (**D**), POD (**E**), GR (**F**) and GST (**G**) in leaves of control, OE-MaGPX3 and OE-MaGPX5 transgenic lines. (**H**) Content of hydrogen peroxide (H_2_O_2_) in leaves of control, OE-MaGPX3 and OE-MaGPX5 transgenic lines. For sampling, the plants were subjected to drought stress by withholding water for 6 days, while the non-stressed plants were grown in normal conditions. Data are means ± SD with three biological replicates (three independent transgenic lines) and each with three technical replicates. Letters indicate significant differences between means, determined using Duncan’s multiple range test (5% α). (**I**) Histochemical staining of hydrogen peroxide (H_2_O_2_) in drought-stressed and non-stressed plants by 3,3-diaminobenzidine (DAB) assay. (**J**) Histochemical staining of superoxide anion (O_2_^−^) in drought-stressed and non-stressed plants by nitrobluetetrazolium (NBT) assay. Three independent T_3_ transgenic lines were used in each case. The presence of H_2_O_2_ and O_2_^−^ were assessed by appearance of brown color and dark blue color after staining with DAB and NBT, respectively. Bars = 1 cm (**I**,**J**).

**Table 1 plants-11-02350-t001:** The differentially expressed proteins (DEPs) of antioxidant system enzymes in mulberry under drought stress.

Protein ID	Protein Description	Fold Change	*p* Value	Regulation	MW [kDa]	Coverage [%]	Unique Peptides	^a^ PSMs	Subcellular Localization
W9S7L1	Glutaredoxin domain-containing protein	2.96	0.000474	Up	13.3890	14.4	2	2	Chloroplast
W9SE23	Peroxidase	2.85	0.000120	Up	36.4880	13.6	4	27	Chloroplast
W9QHE0	Glutathione peroxidase	2.48	0.000983	Up	18.4420	13.4	1	21	Cytoplasm
W9QT41	Glutathione peroxidase	2.27	0.003078	Up	18.9620	38.2	8	20	Mitochondria
W9QH65	Glutathione peroxidase	2.04	0.000421	Up	26.5190	18.6	5	45	Chloroplast
W9RCX8	Glutathione S-transferase	2.02	0.001778	Up	26.7850	48.3	9	74	Cytoplasm
W9RT74	Glutathione peroxidase	1.99	0.000374	Up	20.5060	32.1	5	21	Cytoplasm
W9SDB3	Glutathione peroxidase	1.97	0.000211	Up	26.0130	29.2	5	37	Chloroplast
W9SDD7	Thioredoxin	1.85	0.000336	Up	13.2220	22.7	3	9	Chloroplast
W9QZP8	Glutaredoxin-C5	1.77	0.013193	Up	18.9360	26.1	3	9	Chloroplast
W9SBA6	Glutaredoxin domain-containing protein	1.69	0.001711	Up	15.0310	30.8	3	34	Chloroplast
W9S168	Glutathione S-transferase	1.68	0.000215	Up	39.9490	30.5	11	69	Chloroplast
W9SW93	Thioredoxin-like 3-1	1.67	0.000097	Up	21.2490	11.5	2	3	Cytoplasm
W9RMD9	Thioredoxin-like fold containing protein	1.66	0.000454	Up	35.4970	13	5	17	Vacuolar membrane
W9SCK7	Monothiol glutaredoxin-S16	1.57	0.001539	Up	32.5700	14.4	3	10	Chloroplast
W9QVC2	Peroxiredoxin	1.56	0.004432	Up	22.4790	42.1	7	52	Chloroplast, mitochondria
W9SEV0	Peroxiredoxin	1.54	0.000664	Up	17.2870	36.4	6	44	Cytoplasm
W9R4T1	Thioredoxin O1	1.53	0.002092	Up	28.3650	12.3	4	13	Chloroplast
W9SCK7	Monothiol glutaredoxin-S16	1.57	0.001539	Up	32.5700	14.4	3	10	Chloroplast
W9QVC2	Peroxiredoxin	1.56	0.004432	Up	22.4790	42.1	7	52	Chloroplast, mitochondria
W9SEM5	Ferredoxin-thioredoxin reductase, catalytic chain	1.56	0.002258	Up	16.2540	40	6	14	Chloroplast
W9SEV0	Peroxiredoxin	1.54	0.000664	Up	17.2870	36.4	6	44	Cytoplasm
W9R4T1	Thioredoxin O1	1.53	0.002092	Up	28.3650	12.3	4	13	Chloroplast
W9SVF9	Monothiol glutaredoxin-S7	0.64	0.006054	Down	20.0390	9.8	1	1	Chloroplast
W9RQI7	L-ascorbate oxidase-like protein	0.64	0.007718	Down	60.3300	4.6	2	3	Chloroplast
W9SBU2	SodC protein	0.62	0.005244	Down	29.3360	34.2	6	89	Chloroplast
W9RYX2	Thioredoxin reductase	0.53	0.004546	Down	56.9680	8.1	3	4	Chloroplast

^a^ peptide spectrum matches.

## Data Availability

The iTRAQ-based proteomic dataset has been submitted to ProteomeXchange (http://www.proteomexchange.org/, (accessed on 1 December 2019)) via the PRIDE database (Project accession: PXD010227; reviewer account details: username: reviewer20085@ebi.ac.uk; password: 9RNDiGXZ).

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
