# Peer review of "Quantitative Proteomics and Functional Characterization Reveal That Glutathione Peroxidases Act as Important Antioxidant Regulators in Mulberry Response to Drought Stress"

_plants, 2022, doi:10.3390/plants11182350_

Round 1

Reviewer 1 Report

Manuscript ID:plants-1872810

Title:Quantitative proteomics and functional characterization reveal that glutathione peroxidases act as important antioxidant regulators in drought stress response and tolerance in mulberry

Recommendation: Accepted with minor revision

Comments to authors:

The leaves of mulberry (Morus alba L.) have served as the unique feed for silkworm or sericulture over thousands of years in China, and the trees distributed all over the world. Due to the difficulties in genetic manipulation and transformation, little is known about the molecular mechanism of mulberry in response to drought stresses. In this manuscript, the authors performed quantitative proteomics and identified drought responsive DEPs, the glutathione peroxidases (GPX) isoforms in mulberry. They further functionally characterized the GPX isoforms by genetic overexpression in Arabidopsis. According to proteomic analysis, they identified 604 DEPs between stressed and non-stressed plants. The proteomic profiles associated with cellular redox and antioxidant system were extensively changed in mulberry under drought stress. In fact, this is often the case in other plants. Importantly, this manuscript also indicated GPX proteins represent the most responsive antioxidant enzymes in mulberrys response to drought stress, as five GPX isoforms were up-regulation in a total of 6 GPX isoforms in the mulberry genome. By investigating five GPX isoforms in transgenic Arabidopsis lines, they indicated that increased expression of MaGPX3 and MaGPX5 led to significant enhancements of antioxidant activity and ROS scavenging. Accordingly, the transgenic lines showed higher resistance to drought stress. Based on these results, the authors suggested a positive role of GPX-based antioxidant system in mulberry response to drought stress, and considered that the GPX isoforms can regulate drought tolerance through activating antioxidant system and ROS scavenging. As is indicated in the introduction section, besides its sericultural purpose, mulberry has been planted in arid or semi-arid areas in China for its ecological functions. In this respect, the results of this research will provide an alternative to evaluation of drought tolerant germplasm in mulberry. On the whole, this is a nice manuscript that fit the scope of Plants. It is well organized and written, although the length of this manuscript is too long.

Some minor comments below:

1. P1 L2, I have a doubt regarding the title. Drought stress response and tolerance, which one is more important?

2. P2 L60, this sentence is hard to read. Please pay attention to the usage of word "the" in this sentence and make a concise expression.

3. P2 L94-P3 L103, plant GPXs are non-haeme thiol peroxidases that catalyze the  reduction of H2O2 (or organic hydroperoxides) to water or the respective alcohols using reduced glutathione or thioredoxin . Whats their difference between plant and animal GPX?

4. P4 L152-163, for consistency with figure, the control should be replaced by non-stress in the caption. Fig. 1 should be full name.

5. P6, Fig.2D, the title of vertical axis should be supplied.

6. P9 L293, this sentence The GPX enzymatic activity should by replaced by The enzymatic activity of GPX.

7. P10 L324, for consistency, the verb confer should be used by past tense. It is also the case in L354.

8. P11 L388, the wold the before antioxidant enzymatic activities should be deleted.

9. P16 L629, authors considered the relative levels of GPX genes and enzymatic activity might be considered as an important index of evaluation of drought-tolerant mulberry germplasm. What is your research plan about evaluation of drought-tolerant mulberry?

Author Response

Response to Reviewer 1 Comments

Comments to authors:

The leaves of mulberry (Morus alba L.) have served as the unique feed for silkworm or sericulture over thousands of years in China, and the trees distributed all over the world. Due to the difficulties in genetic manipulation and transformation, little is known about the molecular mechanism of mulberry in response to drought stresses. In this manuscript, the authors performed quantitative proteomics and identified drought responsive DEPs, the glutathione peroxidases (GPX) isoforms in mulberry. They further functionally characterized the GPX isoforms by genetic overexpression in Arabidopsis. According to proteomic analysis, they identified 604 DEPs between stressed and non-stressed plants. The proteomic profiles associated with cellular redox and antioxidant system were extensively changed in mulberry under drought stress. In fact, this is often the case in other plants. Importantly, this manuscript also indicated GPX proteins represent the most responsive antioxidant enzymes in mulberry’s response to drought stress, as five GPX isoforms were up-regulation in a total of 6 GPX isoforms in the mulberry genome. By investigating five GPX isoforms in transgenic Arabidopsis lines, they indicated that increased expression of MaGPX3 and MaGPX5 led to significant enhancements of antioxidant activity and ROS scavenging. Accordingly, the transgenic lines showed higher resistance to drought stress. Based on these results, the authors suggested a positive role of GPX-based antioxidant system in mulberry response to drought stress, and considered that the GPX isoforms can regulate drought tolerance through activating antioxidant system and ROS scavenging. As is indicated in the introduction section, besides its sericultural purpose, mulberry has been planted in arid or semi-arid areas in China for its ecological functions. In this respect, the results of this research will provide an alternative to evaluation of drought tolerant germplasm in mulberry. On the whole, this is a nice manuscript that fit the scope of Plants. It is well organized and written, although the length of this manuscript is too long.

Response: Thank you very much for your consideration on our manuscript. We have made further improvement in the new revised version.

Some minor comments below:

  1. P1 L2, I have a doubt regarding the title. Drought stress response and tolerance, which one is more important?

Response: We have revised the title in this new version.

  1. P2 L60, this sentence is hard to read. Please pay attention to the usage of word "the" in this sentence and make a concise expression.

Response: Revised.

  1. P2 L94-P3 L103, plant GPXs are non-haeme thiol peroxidases that catalyze the  reduction of H2O2 (or organic hydroperoxides) to water or the respective alcohols using reduced glutathione or thioredoxin …. What’s their difference between plant and animal GPX?

Response:  Glutathione peroxidase is a class of thiol-based peroxidases that are ubiquitous in the body and it can use glutathione (GSH) or other reducing equivalents to catalyze hydrogen peroxide, organic peroxides and phospholipid hydroperoxide reduction, so as to protect the cell membrane from reactive oxygen species (ROS) damage, maintain the normal function of cell. In animals, GPX is the core component of antioxidant metabolism, and its active center is selenocysteine. While the active site of plant GPX is cysteine, which is much weaker in scavenging reactive oxygen species than other types of antioxidant enzymes (such as catalase or ascorbate peroxidase), suggesting that plant GPX may have other important functions besides antioxidant activity.

  1. P4 L152-163, for consistency with figure, the “control” should be replaced by “non-stress” in the caption. “Fig. 1”should be full name.

Response: Revised.

  1. P6, Fig.2D, the title of vertical axis should be supplied.

Response: We have added title for this figure.

  1. P9 L293, this sentence “The GPX enzymatic activity” should by replaced by “The enzymatic activity of GPX”.

Response: Revised.

  1. P10 L324, for consistency, the verb “confer” should be used by past tense. It is also the case in L354.

Response: Revised.

  1. P11 L388, the wold “the” before “antioxidant enzymatic activities” should be deleted.

Response: Deleted in revised version.

  1. P16 L629, authors considered the relative levels of GPX genes and enzymatic activity might be considered as an important index of evaluation of drought-tolerant mulberry germplasm. What is your research plan about evaluation of drought-tolerant mulberry?

Response: Based on the consideration that higher abundance of GPX isoforms associated with greater antioxidant abilities and higher tolerance to drought stress in mulberry, we will perform a screen on more than 100 mulberry varieties to analyze GPX genes expression and enzymatic activity in our next research.

Reviewer 2 Report

General comments

I have read the manuscript (Plants -1872810). Entitle: Quantitative proteomics and functional characterization reveal that glutathione peroxidases act as important antioxidant regulators in drought stress response and tolerance in mulberry written by Minjuan Zhang et. al., for publication of plant MDPI. In this study, the author investigated the quantitative proteomics was applied to elucidate the proteomic response to drought stress in mulberry, and differentially expressed proteins (DEPs) were identified via LC-MS/MS. In this study the main results indicated that GPX-based antioxidant system play important role in mulberry response to drought stress, and the GPX isoforms could enhance drought tolerance through activating antioxidant system for ROS scavenging.

The overall research is well conducted, and research is obvious application potential for the readers and manuscript is much valuable. However, I found some points, especially the flow of the text and lack of potential references, and lack of connection of story in different paragraphs, especially in the introduction and discussion sections. The author should provide enough examples and their interpretation of different traits of physiological and biochemical responses by the latest and appropriate references, some of which I mentioned below. Overall after I evaluate and request the author for this manuscript as a “MAJOR REVISION”.

Major suggestions

1)  Introduction: The introduction is well starting with the economic importance of mulberry and drought tolerance which is highly appreciated. However, the important message in the introduction especially the effect of drought to any crop/plant should be mentioned primarily in the introduction section somewhere in the second paragraph. The article DOI:10.1016/j.scienta.2018.11.021 better presented the drought effect on the plant, please follow this article as a reference and mentioned that “drought reduced the morphological and physiological traits, reduce the photosynthesis, leaf water potential and sap movement and reduction of stomatal conductance.

   2) Hypothesis and objectives in the introduction: The author should make more clearly present the research hypothesis first and then only its objectives parts secondly in the last paragraph of the introduction section. The author should be well connected to these two parts while mentioning the research objective. The hypothesis should be very clear in the introduction sections because, without appropriate literature, questions, or hypotheses in the introduction section the entire text will be unclear. 

3) Discussion: Author should Improve the first paragraph of the discussion more logically with potential references because the main theme of antioxidant and secondary metabolites under drought and release the of ROS (why ROS is emerging in stress conditions?). Refer to these articles (1) https://doi.org/10.1038/s41598-019-55889 (2) https://doi.org/10.1016/j.scitotenv.2021.146466 and mention somewhere in that paragraph “abiotic stress especially environmental stress (I.e. drought) plant produces the ROS when these plant exposed to the stress condition and plant produce antioxidant, flavonoids, and secondary metabolites play to the role for protecting the plant for detoxifying ROS and protect the plant to protect the abnormal condition (i.e. stress) and protein and amino acid stabilization”.

   Some other comments

 4) Line no. 152: In this figure “Fig 1C” author shouws that the reduction of chlorophyll significantly but in the discussion I do not see it discussion. Please refer to article https://doi.org/10.1016/j.foreco.2020.118099” and improved the text accordingly. Generally physiological performance specially Pn and Gs increased due to the increase the Chlorophyll content because those help to capture the better light and higher amount of light due to Chl. then higher possibility of Pn because of conversation of light energy change into the chemical energy” generally the drought stress reduce the Chl content of the plant.

5) Line no. 315: It will be great if author consider the figure font in the x axis and Y axis and inside the text of each figure. Please, if possible, apply this in other figures too because the text is very small and not clear.

6) Line no. 367: Is author define somewhere about the definition of relative water content (RWC) inside the manuscript. Please mention even the single line about this somewhere inside the manuscript even in footnote/ or fig title because it is different than the Volumetric water content (VWC).

7) Line no. 468: Author should consider the discussion section seriously; it is not concise well. The text is also not clear specially in the second paragraph. Please make it more concise and use only potential refeence and their description. Specially the Ln 555-607 his paragraph author should further improve.

8) Line no. 723: The author should consider the title of 4.3. I not see any physiological traits inside this subtile. Are not they all the biochemical traits (antioxidant and seciondary metabolities) or enzymatic. Author should consider this.

9) Line no. 806: How author determine the ROS? It is incomplete and not clear author should rephrase it again and add some text to make it clearer.

10) Line no. 816 (Conclusion): The conclusion should not be repetitive in the abstract or a summary of the results section. I would love to read striking points and take-home messages that will linger in the readers’ minds. What is the novelty, how does the study elucidate some questions in this field, and the contributions the paper may offer to the scientific community?

11) Line no. 901 (Reference): please double-check the citations, their style, spell check, and other grammatical errors. moreover, I request to the authors for revision throughout the manuscript according to the journal rules.

 Good Luck!

Author Response

Response to Reviewer 2 Comments

General comments

I have read the manuscript (Plants -1872810). Entitle: Quantitative proteomics and functional characterization reveal that glutathione peroxidases act as important antioxidant regulators in drought stress response and tolerance in mulberry written by Minjuan Zhang et. al., for publication of plant MDPI. In this study, the author investigated the quantitative proteomics was applied to elucidate the proteomic response to drought stress in mulberry, and differentially expressed proteins (DEPs) were identified via LC-MS/MS. In this study the main results indicated that GPX-based antioxidant system play important role in mulberry response to drought stress, and the GPX isoforms could enhance drought tolerance through activating antioxidant system for ROS scavenging.

The overall research is well conducted, and research is obvious application potential for the readers and manuscript is much valuable. However, I found some points, especially the flow of the text and lack of potential references, and lack of connection of story in different paragraphs, especially in the introduction and discussion sections. The author should provide enough examples and their interpretation of different traits of physiological and biochemical responses by the latest and appropriate references, some of which I mentioned below. Overall after I evaluate and request the author for this manuscript as a “MAJOR REVISION”.

Response: Thanks very much for the constructive comments on our manuscript. According to the suggestions, we have made much improvement and revision especially on the introduction and discussion sections. Compared with the old version, this new version is more readable and concise. We also supplied more references citation as reviewer mentioned. We hope this new version will be more acceptable for publication in PLANTS.

Major suggestions

1)  Introduction: The introduction is well starting with the economic importance of mulberry and drought tolerance which is highly appreciated. However, the important message in the introduction especially the effect of drought to any crop/plant should be mentioned primarily in the introduction section somewhere in the second paragraph. The article DOI:10.1016/j.scienta.2018.11.021 better presented the drought effect on the plant, please follow this article as a reference and mentioned that “drought reduced the morphological and physiological traits, reduce the photosynthesis, leaf water potential and sap movement and reduction of stomatal conductance.

Response to point 1: According to the suggestions, we have made careful revision in the whole introduction section. We have strengthen the connection of each paragraph. The effect of drought to any crop/plant was addressed and the article DOI:10.1016/j.scienta.2018.11.021 was citated as reference in this new version.

2) Hypothesis and objectives in the introduction: The author should make more clearly present the research hypothesis first and then only its objectives parts secondly in the last paragraph of the introduction section. The author should be well connected to these two parts while mentioning the research objective. The hypothesis should be very clear in the introduction sections because, without appropriate literature, questions, or hypotheses in the introduction section the entire text will be unclear. 

Response to point 2: According to the suggestions, this paragraph was rewritten. The research hypothesis and its objectives have been clearly present in this new version.

3) Discussion: Author should Improve the first paragraph of the discussion more logically with potential references because the main theme of antioxidant and secondary metabolites under drought and release the of ROS (why ROS is emerging in stress conditions?). Refer to these articles (1) https://doi.org/10.1038/s41598-019-55889 (2) https://doi.org/10.1016/j.scitotenv.2021.146466 and mention somewhere in that paragraph “abiotic stress especially environmental stress (I.e. drought) plant produces the ROS when these plant exposed to the stress condition and plant produce antioxidant, flavonoids, and secondary metabolites play to the role for protecting the plant for detoxifying ROS and protect the plant to protect the abnormal condition (i.e. stress) and protein and amino acid stabilization”.

Response to point 3: The introduction section has been reorganized and reduced. So it will be more concise in this new version. Appropriate references including that recommended by reviewer were supplied in this new version. We have mentioned in the paragraph “Under abiotic stress especially environmental stress (i.e. drought), plant produces more ROS than it needs; however, the plant also produces more antioxidant, flavonoids, and secondary metabolites which play the role for protecting the plant for detoxifying ROS and maintaining protein and amino acid stabilization.” Please refer to Line496.

 Some other comments

 4) Line no. 152: In this figure “Fig 1C” author shouws that the reduction of chlorophyll significantly but in the discussion I do not see it discussion. Please refer to article https://doi.org/10.1016/j.foreco.2020.118099” and improved the text accordingly. Generally physiological performance specially Pn and Gs increased due to the increase the Chlorophyll content because those help to capture the better light and higher amount of light due to Chl. then higher possibility of Pn because of conversation of light energy change into the chemical energy” generally the drought stress reduce the Chl content of the plant.

Response to point 4: We have discussed this result and supplied the article citation in the new version. Please see Line 451-462.

5) Line no. 315: It will be great if author consider the figure font in the x axis and Y axis and inside the text of each figure. Please, if possible, apply this in other figures too because the text is very small and not clear.

Response to point 5: We have reorganized the figure font and figure size for all this manuscript. In this new version, the figure text will be clear and easy for reading.

6) Line no. 367: Is author define somewhere about the definition of relative water content (RWC) inside the manuscript. Please mention even the single line about this somewhere inside the manuscript even in footnote/ or fig title because it is different than the Volumetric water content (VWC).

Response to point 6: In this new version, we have defined relative water content (RWC) and RWL detailly in the methods section.

7) Line no. 468: Author should consider the discussion section seriously; it is not concise well. The text is also not clear specially in the second paragraph. Please make it more concise and use only potential refeence and their description. Specially the Ln 555-607 his paragraph author should further improve.

Response to point 7: The discussion section has been reorganized and improved. Some irrelevant paragraphs have been deleted or reduced. So, it will be more concise in this revised version. Appropriate references including that recommended by reviewer were supplied in this new version.

8) Line no. 723: The author should consider the title of 4.3. I not see any physiological traits inside this subtile. Are not they all the biochemical traits (antioxidant and seciondary metabolities) or enzymatic. Author should consider this.

Response to point 8: Revised (Please see Line 641).

9) Line no. 806: How author determine the ROS? It is incomplete and not clear author should rephrase it again and add some text to make it clearer.

Response to point 9: Revised (Please see Line 733).

10) Line no. 816 (Conclusion): The conclusion should not be repetitive in the abstract or a summary of the results section. I would love to read striking points and take-home messages that will linger in the readers’ minds. What is the novelty, how does the study elucidate some questions in this field, and the contributions the paper may offer to the scientific community?

Response to point 10: This section has been rewritten and improved.

11) Line no. 901 (Reference): please double-check the citations, their style, spell check, and other grammatical errors. moreover, I request to the authors for revision throughout the manuscript according to the journal rules.

Response to point 11: The reference section has been revised and we have made revision for throughout the manuscript to the journal rules.

Round 2

Reviewer 2 Report

Dear Author

I have read the revised manuscript (Plants-1872810). Titled: Quantitative proteomics and functional characterization reveal that glutathione peroxidases act as important antioxidant regulators in drought stress response and tolerance in mulberry for publication of Plants MDPI. This is the second submission made by the author. The author addressed all the questions and suggestions that I raised the issue in the review of the original manuscript. I satisfy the author’s revisions throughout the paper. The author well addresses the abstract issues. Especially author improved the introduction and discussion section very well inflow. Now, this manuscript improved the flow of writing, which was comparatively shallow in the original version but in this revised copy author addressed all the quarries and suggestions very well. Before accepting this manuscript if there is anything needed to be revised by the author, especially English grammar, or spell check, I request this manuscript is currently in “Minor Revision” and any grammatical error author may improve in this stage. Thank you.

Author Response

Response to Reviewer 2

Thanks very much for your commendable review on our manuscript. According to the suggestions, we have made further improvement on the manuscript. The whole manuscript including English grammar and few typing error were revised and checked again carefully. All changes were marked-up in the revised version. We hope these revisions are satisfactory and  the revised version will be acceptable for publication in PLANTS.
Thank you very much for your work concerning our paper.